# “Time Is out of Joint” in Pluripotent Stem Cells: How and Why

**DOI:** 10.3390/ijms25042063

**Published:** 2024-02-08

**Authors:** Francesca Agriesti, Olga Cela, Nazzareno Capitanio

**Affiliations:** Department of Clinical and Experimental Medicine, University of Foggia, 71122 Foggia, Italy; olga.cela@unifg.it (O.C.); nazzareno.capitanio@unifg.it (N.C.)

**Keywords:** circadian rhythm, clock genes, pluripotent stem cells, reprogramming, cellular differentiation

## Abstract

The circadian rhythm is necessary for the homeostasis and health of living organisms. Molecular clocks interconnected by transcription/translation feedback loops exist in most cells of the body. A puzzling exemption to this, otherwise, general biological hallmark is given by the cell physiology of pluripotent stem cells (PSCs) that lack circadian oscillations gradually acquired following their in vivo programmed differentiation. This process can be nicely phenocopied following in vitro commitment and reversed during the reprogramming of somatic cells to induce PSCs. The current understanding of how and why pluripotency is “time-uncoupled” is largely incomplete. A complex picture is emerging where the circadian core clockwork is negatively regulated in PSCs at the post-transcriptional/translational, epigenetic, and other-clock-interaction levels. Moreover, non-canonical functions of circadian core-work components in the balance between pluripotency identity and metabolic-driven cell reprogramming are emerging. This review selects and discusses results of relevant recent investigations providing major insights into this context.

## 1. Introduction

The circadian clock governs the temporal oscillation of biological functions at the molecular level, not only to allow organisms to align with external environmental cycles, such as light/dark cycles, but also to adapt to intrinsic physiological and cell-fate determining processes. The molecular clock machinery, heavily conserved during the course of the evolution, further developed through metabolic and epigenetic regulation, whose signaling pathways are rewired following circadian reprogramming. The dysregulation of circadian homeostasis is hallmarked by the onset of diseases and acceleration of aging [1,2,3].

Studies in recent years have revealed the essential role for biological clocks in regulating stem cell circadian homeostasis [4,5,6,7,8,9]. Importantly, although a molecular circadian clock resides in almost all cells in in vivo and in vitro cultures, cells such as germline, zygote, and pluripotent stem cells (PSCs) show no discernible circadian rhythms of clock gene expression [10,11,12]. These pluripotent cells, however, gradually and autonomously develop circadian gene expression rhythms during embryonic development and in vitro differentiation [13,14]. On the other hand, established circadian rhythms disappear on the reprogramming of differentiated cells into induced PSCs (iPSCs) [12]. These findings suggest that the circadian clock system is subjected to reprogramming parallel to the shift between somatic and pluripotent cellular programs, convincingly raising the possibility that the functional remodeling of the circadian clock system is likely coupled to embryonic development and cellular reprogramming, which may contribute to the sustainment of stem cell differentiation potency and, ultimately, regenerative capacity.

In this review, we describe how and why circadian oscillations are hampered in PSCs, in terms of candidate mechanisms accountable for the lack of rhythmicity and of the biological and physiological significance of the delayed emergence of circadian clock oscillations observed during cellular differentiation. In addition, new insights about non-canonical functions of circadian factors in the balance between pluripotency identity and metabolic-driven cell reprogramming are also reviewed.

## 2. Hierarchical Organization of Circadian Clock Network

In mammals, the circadian clock comprises a central clock located in the hypothalamic superchiasmatic nucleus (SNC) and peripheral clocks that are present in practically all organs of the body and cells. The SNC is mainly entrained by light signals form the retina and, subsequently, synchronizes the peripheral clocks through neuronal and humoral factors [15]. Interestingly, when ablating the SCN, these peripheral clocks remain functional, and they synchronized with feeding regime and behavioural rhythms [16,17,18]. Most notably, even under culturing conditions, cells display circadian profiles in given functions once in vitro synchronization protocols have been adopted [19,20,21,22], thus demonstrating the occurrence of cellular autonomous circadian time-keeping capacity.

Mechanistically, the cellular circadian clock oscillation is driven by interlocked positive and negative transcription/translational feed-back loops (TTFLs) carried into effect by a set of core clock genes (Figure 1a). Fundamental for the TTFLs are two basic-helix-loop-helix (bHLH) PAS transcription factors, Brain and Muscle ARNT-Like 1 (BMAL1) and Circadian Locomotor Output Cycles Kaput (CLOCK), that heterodimerize and transactivate the expression of their own inhibitors, PERs (encoded by *Per 1/2/3*) and CRYs (encoded by *Cryptocrome 1/2*). PER and CRY proteins accumulate in the cytoplasm, and, if not degraded, they can dimerize and shuttle in the nucleus where they promote the dissociation of the BMAL-CLOCK heterodimers from the E-box element on the DNA promoter regions [23,24]. The specific DNA sequence of the E-box motif is CANNTG (where N can be any nucleotide) with a palindromic canonical sequence of CACGTG [25].

In a secondary feed-back loop, the BMAL1-CLOCK complex drives the transcription of RORα/β and REV-ERBα/β (encoded by *Rorα/β* and *Nr1d1/2*, respectively), which activate and inhibit *Bmal1* expression, respectively, competing for its RORE (retinoic acid-related orphan receptor binding element) promoter [26,27,28].

A third and last transcriptional loop comprises the D-box binding protein (DBP), hepatic leukemia factor (HLF), and thyrotrophic embryonic factor (TEF), contributing to positive regulation. In contrast, nuclear factor interleukin 3 regulated (NFIL3) plays a role in negative regulation [29]. While this supplementary loop may not be essential for circadian oscillations, it enhances robustness and precision in determining the period. 

In addition to the intricate transcriptional and translational control system, circadian rhythmicity is influenced by covalent post-translational modifications (PTMs) of the core protein clockwork. It is also regulated by epigenetic mechanisms, including DNA methylation at CpG islands, non-coding RNAs, and post-translational modifications of histones. These various epigenetic mechanisms have been associated with the initiation and precise adjustment of circadian rhythmicity in gene expression [7].

Collectively, this core transcriptional, translational network evolved in such a way that the delay in the gene expression of the clock components fits the period of the earth rotation. Most notably, down-stream of the aforementioned TTFLs is the time-controlled expression of an unexpected large number of genes (CGGs, clock-controlled genes) that share similar binding sites with the core clock transcription factors on their promoter [24].

Although the core pathway is common across tissues, the resulting rhythmic transcription of clock-controlled genes (CCGs) is markedly tissue-specific. This specificity is crucial to fulfill the physiological requirements of each organ. Studies indicate that up to 43% (in rodents [30]) or 82% (in primates [31]) of all protein-coding genes exhibit circadian expression in at least one organ.

The significance of preserving the proper functioning of the biological clock is underscored by its involvement in various pathological conditions, including disrupted metabolism, cardiovascular diseases, sleep disorders, cancer [32], neurodegenerative diseases [33], and even hampered regenerative capacities [7]. Therefore, the circadian clock is under intense investigation in differentiated cells as well as in adult stem cells, and even in embryonic cells and PSCs. 

Intriguingly, clock genes-mediated circadian oscillations are remarkably dampened in PSCs and gradually develop following differentiation (Figure 1b). The mechanisms underlying this exquisite property of the pluripotency state are detailed in the next paragraphs.

It should be considered that in addition to the circadian clock, other rhythmic processes have been emerging in somatic cells, although poorly characterized [34]. These are collectively defined as ultradian rhythms with periods of about 12, 8, or 6 h (or even shorter) that are intriguingly secondary harmonics of the primary circadian oscillation. In some cases, ultradian oscillations can be tracked back to circadian oscillations of two or more transcription factors but in an anti-phase [35,36]. In other cases, BMAL1/CLOCK-independent clockworks appear to be involved [37,38,39]. Although, interestingly, a paucity of information is available about these ultradian rhythms in PSCs; therefore, they are not mentioned in this review, with an exception discussed ahead in another paragraph.

## 3. Pluripotent Stem Cells: What Is Known and What Is Still Missing

PSCs, such as embryonic stem cells (ESCs) and induced pluripotent stem cells (iPSCs), are characterized by their ability to indefinitely self-renew and, according to appropriate clues, differentiate into virtually all types of organismal cells. 

ESCs derived from the inner mass of a developing embryo at the blastocyst stage [40,41,42] can proliferate indefinitely in vitro and can give rise to derivatives of all the three germ layers (ecto, meso-, and endoderm); and they can also differentiate into clinically relevant cell types such ad neurons, hepatocyte, and cardiac cells [43] (Figure 2). However, human ESCs-related research is ethically controversial because it involves the destruction of human embryo.

In 2007, a major technological breakthrough in science and medicine was made with the report that human adult cells can be reprogrammed into iPSCs via the overexpression of four transcription factors—*Oct4*, *Sox2*, *Klf4*, and *Myc*—termed “Yamanaka factors” [44,45]. Once generated, iPSCs have the same properties of ESCs, thereby replacing their controversial utilization with a substantial impact on both the basic and clinical levels of biomedical research [46,47]. Shortly after their technological advancement, human iPSCs were promptly employed to create models representing human diseases, often referred to as “disease in a dish”. These models offer several advantages, such as their human origin, ease of accessibility, expandability, capability to differentiate into almost any desired cell type, and the potential for developing personalized medicine using patient-specific iPSCs. Moreover, recent progress in gene-editing technologies, particularly the CRISPR-Cas9 technology, is facilitating the swift generation of genetically defined human iPSC-based disease models [48,49,50,51].

Despite the development of numerous methods for generating human iPSCs, significant variations exist among them in terms of efficiency, quality, speed, cost, and robustness. Various non-integrating techniques are currently in use, employing episomal DNAs [52,53], Sendai virus (SeV) [54], adenovirus (AdVs) [55], Piggy-Bac transposons [56], minicircles [57], synthesized RNAs [58], and recombinant proteins [59] to deliver reprogramming factors into target cells. Episomal DNAs, synthetic mRNAs, and SeV are particularly common among these approaches, as they are frequently employed to generate integration-free iPSCs due to their simplicity and high efficiency. Additionally, numerous enhancers for somatic cell reprogramming have been identified and utilized, either in conjunction with or independently of these technologies [60]. 

Traditional reprogramming methods result in the production of primed iPSCs, which closely resemble PSCs derived from post-implantation embryos. However, it is worth noting that human PSCs can also be identified in pre-implantation embryos, and these PSCs, known as naïve PSCs, arising early in embryonic development are physiologically distinct from primed iPSCs [61,62]. Recently, an improved Sendai viral system for the reprogramming to naïve pluripotency was developed [63]. The naïve iPSCs generated using this method have greater potential to differentiate in vitro into an extra-embryonic cell type [64] than those derived using conventional methods. Thus, the development of optimized protocols to derive naïve PSCs is not only important for research on early human development but can expand the application of PSCs in regenerative medicine. 

Ensuring the quality and safety of iPSCs requires a comprehensive understanding of the mechanisms driving the reprogramming of somatic cells. This process involves significant alterations in gene expression, metabolic and epigenetic statuses, as well as cellular structure and functions [55].

Unfortunately, the biology of these cells has not been fully elucidated enough, and basic knowledge about important functions, such as the circadian clock machinery, is still incomplete. When PSCs are used for direct differentiation mimicking embryonic development, it is important to clarify the impact of differentiation agents on the circadian network machinery since some of them, such as forskolin and dexamethasone, often employed for deriving naïve iPSCs and mesenchymal stem cell-like cells [65,66] and for specific lineage commitment [67,68,69], respectively, are compounds commonly used as in vitro intrinsic-clock synchronizers. 

Notably, a recent study reported that clock genes do not respond to the synchronizing agents in iPSCs; instead, a unique circadian-like rhythm is induced by the temperature rhythm, which is likely due to a feedback loop via the hypoxia inducible factor 1α (HIF-1α) rather than heat or cold inducible proteins and clock genes [70]. To this regard, it is worth noting a reciprocal regulation between the circadian clock and the hypoxia signaling at the genome level in mammals. Accordingly, oxygen has been identified as a cue for the entrainment of molecular clocks that are synchronized in a HIF-1α-dependent manner. In particular, HIF-1α and BMAL1 can heterodimerize and typically bind to one or the other of the promoter/enhancer regions, E-box, and hypoxia response element (HRE) because of overlapping DNA sequences. Thus, HIF-1α may be capable of binding to the E-box of some circadian genes regulating their expression [71]. Noteworthy, oxygen metabolism is also able to shape and influence the cell fate of adult stem cells [72].

Furthermore, it has been established that the circadian clock genes play a crucial role in governing the cell cycle of mouse ESCs (mESCs) [73] whose state is known to affect both the differentiation capacity and the proliferation rate [74,75]. In particular, it has been reported that the knockout of *Clock* in mESCs slows down the cell cycle by decreasing the expression levels of C-Myc, CyclinD1, CDK1, CDK2, and PCNA. In addition, enhanced expression of apoptosis-related markers such as Bax, Bcl-2, caspase 3, and caspase 9 was observed. These combined effects would influence the proliferation rate in *Clock*-deficient mESCs. Likewise, in human embryonic development, it has been reported that there exists a complex reciprocal interplay between the circadian clock TTFLs and factors controlling the G1/S and G2/M cell cycle checkpoints [5]. Noticible, among those interations that involving BMAL1/CLOCK and cMyc, one of the “Yamanka factors” is of relevance (see above). 

## 4. Circadian Rhythm in Pluripotent Stem Cells: The Reasons for Silence

By using bioluminescent reporter systems and extensive time-resolved analyses of clock gene transcripts, several reports have clearly shown in PSCs, murine, and human ESCs and iPSCs that there is an absence of circadian rhythms associated with clock gene expression [11,12,13,14,76]. Nevertheless, most of the core clock genes (*Per1*, *Per2*, *Clock*, *Bmal1*, *Cry1* and *Cry2*) were found to be expressed in these cells, posing the question of how and why circadian oscillations are hampered in PSCs. 

It should be noted that differences in the stoichiometry of core clock gene transcripts between oscillating differentiated cells and undifferentiated cells has been reported with lower expression levels in PSCs [12,13,77]. Given that the expression ratio of core genes and the availability of clock proteins are the primary mechanisms for establishing and maintaining the diurnal oscillatory network, it is reasonable to anticipate that these changes might contribute to the lack of a functionally synchronized clockwork. Possibly, the altered clock factor stoichiometry might shift the role of core clock genes toward other functions such as modulation of proliferation.

In addition to TTFLs, the circadian clock is further regulated by multiple post-translational modifications (PTMs), including phosphorylation, ubiquitination, acetylation, and SUMOylation. These modifications, practically reported for all the components of the core clockwork, affect their interactions, cellular localization, and time-life [78]. The aberrant localization of core clock proteins might contribute to a non-functional clock in PSC cells. PERs proteins in mouse PSCs such as ESCs, iPSCs, and multipotent germline stem cells (mGSCs), are exclusively localized to the cytoplasm [76]. Within normal differentiated cells, PERs and CRYs exhibit dynamic shuttling between the nucleus and cytoplasm, even though their primary localization is in the nucleus. Certain modifications, such as phosphorylation and ubiquitination, of these negative regulators influence their subcellular dynamics and protein stability [79,80,81]. PER2 is phosphorylated on multiple sites by CK1ε/δ and GSK3β; that by a phosphoswitch mechanism of CK1ε/δ leads PER2 to either degradation or nuclear stabilization, thereby changing its subcellular localization [82,83,84]. 

Moreover, it has been proposed that the subtype α2 of the importin family, encoded by *Kpna2*, plays a role in the subcellular localization of PER1/2 proteins. Importin α2 is a nuclear transporter that, together with its partner importin β1 (importin α2/β1), shuttles specific pluripotency factors, such as OCT3/4, but not differentiation-related factors like OCT6 (requiring importin α1/β1), which remain out of the nucleus in mouse ESCs, thereby contributing to the retention of their pluripotent state [85] (Figure 3).

Notably, this nuclear transporter also induces the retention of core clock factors, PER1 and PER2, in the cytoplasm, therefore avoiding the proper nuclear function of the negative feedback loop required for cyclic circadian regulation [77]. In differentiated circadian rhythm-competent cells, the nuclear translocation of PER/CRY is specifically mediated by the importin β KPNB1 independently of the importin α partner [86] (Figure 3). Overall, this evidence would suggest a coordinated trafficking of clock- and stemness-related factors from cytosol to the nucleus in order to maintain the pluripotency state that changes upon commitment. Another key regulatory mechanism that could explain the silencing of circadian oscillations in PSCs, not at the level of translation and/or protein modification, is the post-transcriptional control of CLOCK protein. The CLOCK protein is not found, despite the expression of its mRNA, in mouse ESCs and human iPSCs. Of note, the loss of the Dicer/DGCR8 complex, known to be involved in the biogenesis of microRNA, is linked to the emergence of circadian clock oscillation during development, indicating post-transcriptional regulation of CLOCK mRNA (Figure 4a) [77].

Although NPAS2, a paralogue of CLOCK, can compensate for CLOCK dysfunction [87,88,89], its expression level in undifferentiated human iPSCs and ESCs is extremely lower than that of CLOCK [90], which is instead similar to that observed in mouse ESCs and early embryos [77]. Therefore, the post-transcriptional suppression of CLOCK is considered to be one of the reasons for the lack of a circadian oscillator in undifferentiated human iPSCs.

## 5. Differentiation-Coupled Circadian Clock Development from Pluripotent Stem Cells: Moving on the Road 

Despite the fact that PSCs do not have TTFLs-mediated discernible circadian molecular oscillations, the emergence of robust circadian oscillation is observed to develop gradually and cell-autonomously during in vitro differentiation, as well as during the development in mammals. Accordingly, PSCs differentiation in cultures recapitulates this process, whereas the reprogramming of somatic differentiated cells into iPSCs reverses it (Figure 1b). Hence, the circadian clock development is tightly coupled with the cellular differentiation state [10,12].

The temporal correlation between the gradual elevation in CLOCK protein expression and the robustness of circadian gene expression rhythms offers valuable insights into the mechanisms governing circadian clock development during differentiation. While the expression of CLOCK alone is not adequate for circadian clock oscillation in undifferentiated PSCs, the regulation of CLOCK expression may influence the timing of circadian clock oscillation onset during cellular differentiation and developmental processes in mammals. As a result, a two-step program has been suggested for cellular differentiation-coupled clock development, involving a lineage-dependent cellular commitment followed by the subsequent establishment of transcription-translation feedback loops (TTFLs) of the mammalian circadian clock, with post-transcriptional regulation of the clock acting as a rate-modulating mechanism [77]. These sequential mechanisms may, at least in part, elucidate the delayed emergence of mammalian circadian clock oscillation in the developmental process.

Typically, the epigenetic repression of developmental genes, whether in a steady state or in response to stimuli, occurs in PSCs. DNA methyltransferase 1 (DNMT1) serves as a crucial regulator of differentiation in diverse stem cell types, ensuring global DNA methylation. The absence of DNMT1, as seen in DNMT1 deficiency, disrupts the differentiation-associated development of the circadian clock [10,12]. Thus, it is likely that profound changes in epigenetic landscape, as well as in transcriptome after exit from pluripotency, are a prerequisite for the formation of the functional circadian clock.

Among the various epigenetic modifications, histone H3 lysine 27 trimethylation (H3K27me3), a prominent repressive epigenetic marker and a characteristic feature of facultative heterochromatin [91], has been observed to exert a counteractive influence in the circadian regulation of gene expression. For instance, the *Per1* promoter exhibits rhythmic H3K27me3 marks, facilitated by the histone-lysine N-methyltransferase enhancer of zeste homologue 2 (EZH2), the enzymatic component responsible for the functionality of the polycomb repressive complex 2 (PRC2) [92]. In PSCs, H3K27me3 is located at the promoters of many important developmental regulators [93], and the regulation of clock genes by PRC2 has been recently examined in a study reporting, for the first time, its involvement in the epigenetic regulation of PER1 in human iPSCs (Figure 4b). It has been reported that the epigenetic repression of clock genes by the histone modification of H3K27me3 along with low levels of BMAL1 suppresses the emergence of circadian rhythms in iPSCs. Accordingly, a significant circadian rhythm of clock genes was induced following artificial BMAL1 overexpression and EZH2 inhibition with GSK126 treatment, thus suggesting a new candidate mechanism for the lack of rhythmicity of clock gene expression in iPSCs [94].

Numerous studies have conducted a comparative assessment of BMAL1 expression levels in PSCs and their differentiated counterparts. Ameneiro et al. examined BMAL1 mRNA and protein levels, revealing higher expression in mouse embryonic stem cells (ESCs) compared to mouse embryonic fibroblasts [95]. Additionally, Gallardo et al. conducted a comparison between mouse ESCs and their differentiated neural stem cells (NSCs), finding similar BMAL1 mRNA and protein levels, signifying functional expression in mouse ESCs [96]. Regarding human PSCs, Thakur et al. demonstrated the expression of BMAL1 mRNA in human embryonic stem cells (ESCs), with levels remaining unchanged during spontaneous differentiation [97]. These findings are inconsistent with results reported for human iPSCs by Kaneko at al. and other studies [13,94], leading to the conclusion that lower expression levels of BMAL1 in iPSCs might be insufficient for sustaining/maintaining the circadian clock. These apparently conflicting results could derive from some subtle differences between ESCs and iPSCs.

Collectively, the sequential progression from pluripotency to the initiation of cellular differentiation, coupled with epigenetic alterations, facilitates the precise spatiotemporal expression of clock component proteins such as PER1, BMAL1, and CLOCK. These proteins are essential for the emergence of circadian clock oscillations. Interestingly, human iPSCs necessitate a three- to four-fold-longer differentiation period compared to mouse embryonic stem cells (ESCs)/iPSCs to establish circadian oscillations of gene expression. This difference may potentially reflect the variances in gestation periods between mice and humans, although further investigations are needed to confirm this hypothesis [14].

## 6. Emergence of Ultradian Circadian Oscillations during Ontogenic Differentiation 

The circadian clock is essential for regulating the temporal order of physiological functions of cells and whole organism. Hence, elucidating the molecular basis for the precise developmental timing of the circadian clock’s emergence is not only critical for understanding embryonic development in mammals, as well as cellular differentiation, but also critical for optimizing protocols for efficient cell therapies. The evidence that the CLOCK/BMAL1-mediated TTFLs are dysfunctional in no rhythmic cells would suggest that circadian oscillation is disadvantageous in the early developmental stages. 

While the complete biological significance behind regulating the suppression of the circadian clock and its delayed onset remains not entirely understood in mammals, insights from the fruit fly, *Drosophila melanogaster*, reveal that CLOCK overexpression leads to developmental lethality [98]. Furthermore, post-transcriptional regulation of CLOCK is crucial for proper development [99]. These findings imply that the regulation of CLOCK expression is vital not only for the emergence of the circadian clock but also for the overall developmental process.

New insights in mammals have been provided by a recent study focusing on the interaction between circadian key components and the segmentation clock, another cell-autonomous oscillator, located in the posterior presomitic mesoderm (PMS), which controls somitogenesis and is essential for an intact developmental process in the early developmental stage [100,101]. Umemura et al., in mouse embryonic organoids, demonstrated that the premature expression of CLOCK/BMAL1 proteins significantly impairs the ultradian rhythm of the segmentation clock [102]. In addition, RNA seq analysis uncovered that CLOCK/BMAL1 influences *Hes7* transcription, a crucial bHLH transcriptional factor that inhibits its own expression and oscillates through a negative feedback loop with a period of 2–3 h in mice and 4–5 h in humans [103,104]. Therefore, the premature expression of CLOCK/BMAL1 downregulates *Hes7* gene expression and its LFNG- and Notch-related regulatory pathway [102]. Consequently, the expression of functional CLOCK/BMAL1 severely interferes with the ultradian rhythm of segmentation clock in induced PMS and gastruloids (Figure 5).

Given that the transcriptional activation of *Clock/Bmal1* is crucial for circadian regulatory networks, recent discoveries suggest that the complete suppression of circadian molecular oscillatory mechanisms in early-stage embryos might be necessary during somitogenesis for the proper developmental process in mammals. Thus, the delayed onset of the circadian clock oscillation observed in mammalian development may hold biological and physiological significance.

## 7. Non-Canonical Role of Circadian Factors in Pluripotency and Metabolic-Driven Reprogramming 

PSCs do not possess a canonical TTFLs-based circadian clock; nevertheless, they do express most of the clock factors, though at different levels than somatic cells. Whether they exert specific roles in stem cell maintenance is not fully elucidate. The conditional knockout of *Bmal1* in adult mice has been reported to result in no dramatic phenotypic changes as compared the wild type [105]. This conflicts with the results attained in conventional *Bmal1* knockout mice, which showed accelerated aging and shortening of the life span [106]. A possible explanation for these apparently contradicting results is that BMAL1 displays properties in the developmental time-window that are independent of its role in the clock but influence the later adult life. Hence, clock-related factors may exert off-target functions in proliferative PSCs that are distinct from their circadian and cell division clock-related role in differentiated cells. Alternatively, the transcriptional function of specific core clock proteins acting on hundreds of genes might control, directly or indirectly, portions of the PSCs transcriptome but not in a circadian TTFLs-context. 

As previously discussed, numerous studies have supported the involvement of clock genes in organ development. Gallard et al. demonstrated that *Bmal1* knockout mouse embryonic stem cells (ESCs) exhibit impaired capacity for multi-lineage cell differentiation [96]. Additionally, Ameneiro et al. showed that *Bmal1* depletion leads to the dysregulation of transcriptional programs associated with cell differentiation commitment and disrupts gastrulation in vitro. Interestingly, BMAL1 was found to be dispensable for maintaining the pluripotent state in mouse ESCs [95].

Regarding the impact of BMAL1 on the expression of pluripotency genes in human PSCs, conflicting findings have been reported, possibly due to differences in the primed and naïve states of human and mouse PSCs. Kaneko et al. observed significantly higher *Nanog* mRNA levels in BMAL1-overexpressing iPSCs [94]. In contrast, Gallardo et al. described a significant increase in both mRNA and protein levels of NANOG in *Bmal1* knockout ESCs, while Ameneiro et al. found no significant effect on these levels in *Bmal1* knockout ESCs [95,96].

A recent investigation has shed light on the impact of BMAL1 on the expression of markers associated with both naïve and primed states in iPSCs. The study revealed that BMAL1 overexpression led to an elevation in the levels of naïve markers such as KLF4 and DNMT3L. This finding suggests a potential involvement of BMAL1 in the transition from a primed to a naïve state in PSCs, given that KLF4 plays a crucial role in the conversion of human PSCs into a naïve pluripotent state and in their subsequent maintenance [107,108]. Thus, BMAL1 has been proposed as a novel regulator of pluripotent biology and cell differentiation [96].

Overall, these findings unveil the existence of an uncharacterized function of the molecular clock, which does not rely on the canonical oscillator production of gene transcripts, but which is essential for pluripotency execution and proper embryo development in mammals. 

Mechanistically, the depletion of BMAL1 induced a shift in the basal metabolism of pluripotent cells involving a reduction of glycolysis and an increase in OXPHOS activity accompanied by augmented production mitochondrial reactive oxygen species (ROS) [13] (Figure 6). Alterations in metabolic activity are intricately connected to the transition from pluripotency, partly due to their influence on the epigenome during cell commitment. This, in turn, can regulate pluripotency, differentiation, and somatic cell reprogramming [109,110]. Consistent with this observation, genome-wide immunoprecipitation-based techniques have revealed that BMAL1 targets genes associated with cellular metabolism in somatic cells [111,112,113], suggesting a potential direct regulation of metabolic genes by BMAL1 through chromatin binding.

Collectively, the novel function of BMAL1 in the metabolic control for cell fate determination provides evidence implicating non-canonical circadian clock regulation in development and disease. A metabolic shift occurs throughout the reprogramming process. Broadly, metabolism influences the effectiveness of reprogramming, and, conversely, changes in reprogramming efficiency are often associated with alterations in metabolism [114]. While recent studies have elucidated the overall pattern of metabolic shifts during reprogramming, analyzing metabolic regulation remains challenging. This difficulty arises because the metabolic system is intricately connected to complex networks of feedback and feedforward loops, both at the transcriptional and post-transcriptional levels, including TTFLs, to uphold homeostasis.

Circadian-clock-controlled signaling pathways are rewired by specific metabolic conditions, leading to the creation of new signal-transduction networks [115,116]. The restructuring of the circadian clock system is likely intertwined with embryonic development and cellular reprogramming, potentially contributing to the preservation of stem cell differentiation potency and, ultimately, regenerative capacity. However, it remains unclear whether metabolic features unique to PSCs are interconnected with clock functions. PSCs exhibit a proliferative metabolism involving aerobic glycolysis, glutamine oxidation, and lipid and nucleotide synthesis, in which clock-mediated fine-tuning has been suggested [117,118,119]. A very recent study revealed non-canonical regulatory roles of the circadian clock CRY1 in PSC identity and cellular reprogramming and metabolism [120]. Previous studies also uncovered non-canonical functions of CRY1, as a positive regulator of pluripotent programs, in relation to tissue regeneration and stem cell functions [121,122,123,124]. Consequently, CRYs exhibit a range of versatile and diverse functions by adjusting their quantity in a circadian manner and/or in a stage-specific manner during development, as well as through binding with various molecules. Physiologically, the CRY1 protein is notably abundant during the biological day (active phase), with its expression peaking during the biological night. The circadian oscillation of CRY1 levels plays a role in cyclically regulating various biological functions, such as DNA repair [125,126] and metabolism [127,128] along with the core circadian machinery. 

In their last report, Sato et al. showed that a non-oscillatory clock represents circadian reprogramming in PSCs. In this scenario, pluripotent metabolic signature, such as the activation of SREBP1 and inhibition of AMPK, contributes to the cellular accumulation of CRY1 to dictate pluripotent programs, including self-renewal capacity, maintenance of undifferentiated state, and metabolic programs unique to PSCs. In addition, it has been reported that CRY1 deficiency resulted in altered gene responses to iPSC reprogramming and impaired iPSCs reprogramming efficiency, thus confirming a novel role of CRY1 in promoting iPSCs reprograming (Figure 6) [120].

Overall, these the findings point to CRY1 as a potential molecular regulator of PSCs homeostasis that could contribute to the rheostat of circadian rhythmicity during cellular differentiation and reprogramming, thereby dictating PSCs identity. 

## 8. Conclusions

The present review discussed the importance of the circadian rhythm in the context of PSCs biology. Circadian rhythms drive stem cell metabolism, self-renewal, and differentiation, and can even create stem cell heterogeneity in one tissue to protect the organism from stem cell depletion upon activation. Conversely, the absence of the circadian clock during early development might be necessary for the successful progression of developmental programs, including stem cell expansion, raising the possibility that clock function is actively suppressed during early ontogenesis for resistance to undesired intrinsic and/or environmental circadian cues.

Accordingly, the reason why the circadian rhythmicity is dampened in the stem cell compartments might depend on the simple fact that stem cells need to be unresponsive to certain organismal clues to maintain their undifferentiated status. Indeed, evidence reported in literature clearly shows that in differentiated cells, circadian oscillations are tissue/organ specific both in terms of proteomic, transcriptomic, and metabolomic profiles and phasing/entrainment. This would optimize the inter-organs cross talk at the organismal level to better face the environmental and internal circadian changes. Thus, a full activation of the circadian clock machinery in stem cells would result in undesired commitment to align with the clock of a given lineage, thereby causing exhaustion of the stem cells’ reservoirs. 

Although the biological significance as well as molecular mechanisms of the silenced circadian clockwork in PSC remains an important question to be fully uncovered, it is likely that circadian rhythms will influence cell-based regenerative therapy. Recent progression in developing pharmacologic compounds able to stimulate or inhibit specific components of the circadian clock TTFLs paves the way for novel bio-medical applications [129,130,131]. Targeting the clock in adult stem cells in vivo might enhance tissue regeneration after damage. Another route for treatment could be ex vivo culturing of stem cells to synchronize their clocks and administering them in the time window in which the patient is most receptive and in which the cells are most likely to engraft. 

Among the many physiological alterations occurring during aging, the exhaustion of the adult stem cell reservoir and dysregulation of circadian rhythms are emerging [132,133,134]. Prospectively, deepening the currently appearing complex interplay between the cellular biology of stem cells and rhythmic time-keeping mechanisms may offer new opportunities for healthy and longer life. 

Directing the differentiation of human PSCs toward specific lineages offers an inexhaustible supply of cells for potential therapeutic applications. Moreover, this approach enables the identification of clock-controlled genes in specific cell types that would otherwise be challenging to investigate in humans. Moreover, the optimal use of the knowledge on circadian rhythms and potentially modifying circadian rhythms or clock components could enhance stem cells differentiation and the effect of stem-cell-based regenerative medicine. Revealing the oscillatory networks associated with both cell survival and factor secretion has the potential to enhance cell-based therapy following injuries.

A further aspect deliberately left aside in this review is the circadian clock homeostasis in the context of the cancer stem cells (CSCs). Circadian timing could play a role during the differentiation of CSCs or dedifferentiation of mature cancer cells. Several lines of evidence suggest that a disorganized or hampered circadian clockwork is a risk factor for cancer development, growth, and progression, although it remains controversial if it is a sufficient causal factor on its own [135,136]. The genetic diversity, mutations, tumor promoters, chronic inflammation and immune background, and micro-environmental conditioning hallmark the heterogeneity of the oncological diseases, thus adding a further level of complexity to the understanding of the interplay between the circadian oscillators and the biology of CSCs. Intensive research in this direction is needed as it holds promise for the development of novel and more efficient therapeutic strategies.

## Figures and Tables

**Figure 1 ijms-25-02063-f001:**
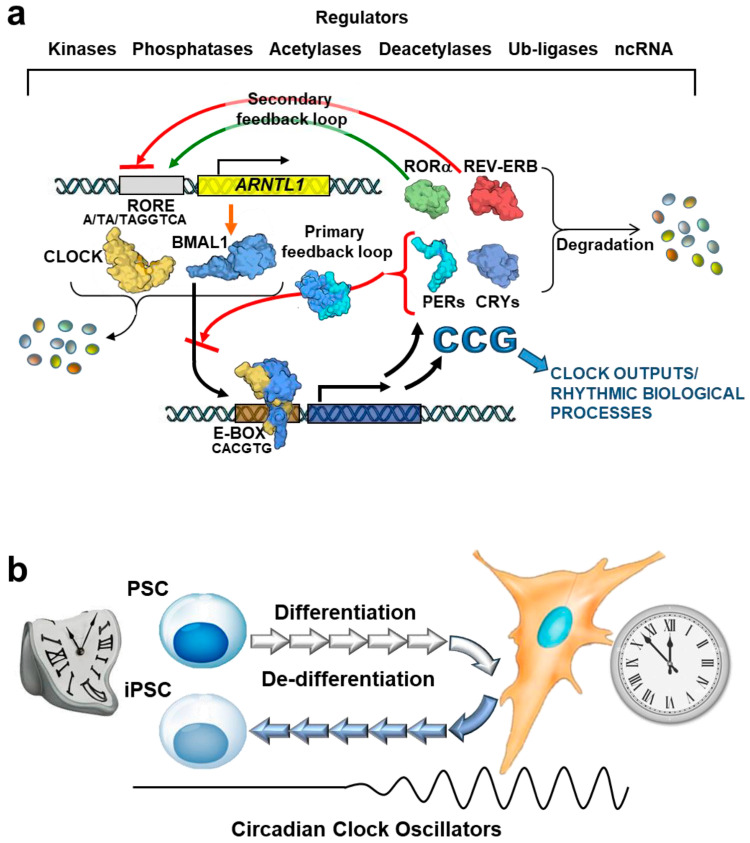
The circadian clock function is coupled to cellular differentiation. (**a**) Schematic representation of the transcriptional/translational feedback loops (TFFLs) of the circadian clock pathway. The transcription factors BMAL1 and CLOCK binds to E-boxes and drive the expression of clock-controlled genes (CCG) and their own inhibitors, PER1 and CRY1, which, if not degraded, block BMAL1::CLOCK transcriptional activity in a primary feedback loop. The ROR and REV-ERVB transcription factors govern the second feedback loop dependent on BMAL1::CLOCK. Through competitive binding to the ROR/REV-ERB-response element (RORE) in regulatory sequences, their proteins activate or repress *Bmal1* transcription. The pathway’s robustness is further influenced by post-transcriptional, translational, and epigenetic modifications, ensuring the establishment of approximately 24 h rhythmic cycles of BMAL1::CLOCK-mediated transcriptional activation in CGCs. (**b**) Emergence of the circadian clock during differentiation. The core TTFLs of the circadian molecular oscillation in PSCs are not detectable but exit from pluripotency, and subsequent commitment of PSCs induces a cell-autonomous robust circadian oscillation that disappears after reprogramming differentiated cells into induced pluripotent cells (iPSCs). The diagram of the proteins shown is from the respective PDB depository code and created with BioRender.com. Ub-ligase, ubiquitin ligase; ncRNA, non-coding RNA.

**Figure 2 ijms-25-02063-f002:**
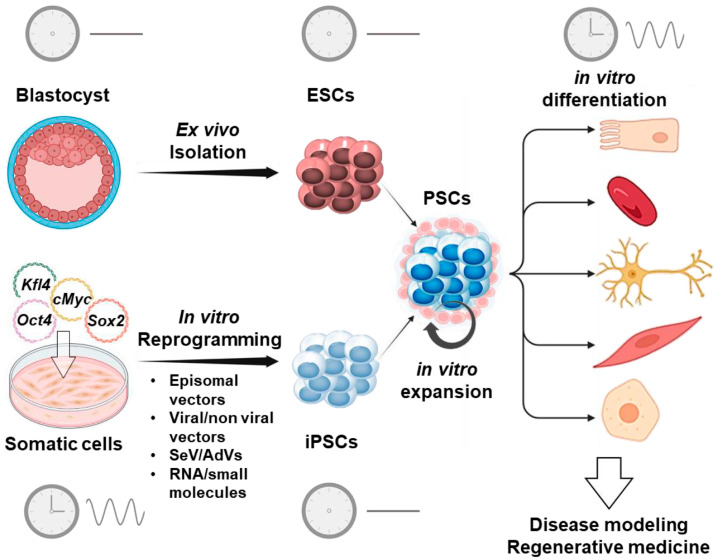
Generation of pluripotent stem cells. The diagram compares the derivation of embryonic stem cell lines (ESCs) from the inner cell mass of the blastocyst and how iPSCs are derived from somatic cells following induction of the “Yamanaka factors” (*Oct4*, *Kfl4*, *Sox2*, *cMyc*) by different reprogramming methods. These PSCs can be expanded indefinitely and then be directed to differentiate in vitro into clinically relevant cell types. Cells differentiated from PSCs are expected to contribute to disease modelling in vitro and to regenerative medicine as cell therapies. The icons of the clock without and with hands imply the absence or presence of circadian oscillators, respectively, in the different cell types shown. SeV, Sendai virus; AdVs, adenovirus.

**Figure 3 ijms-25-02063-f003:**
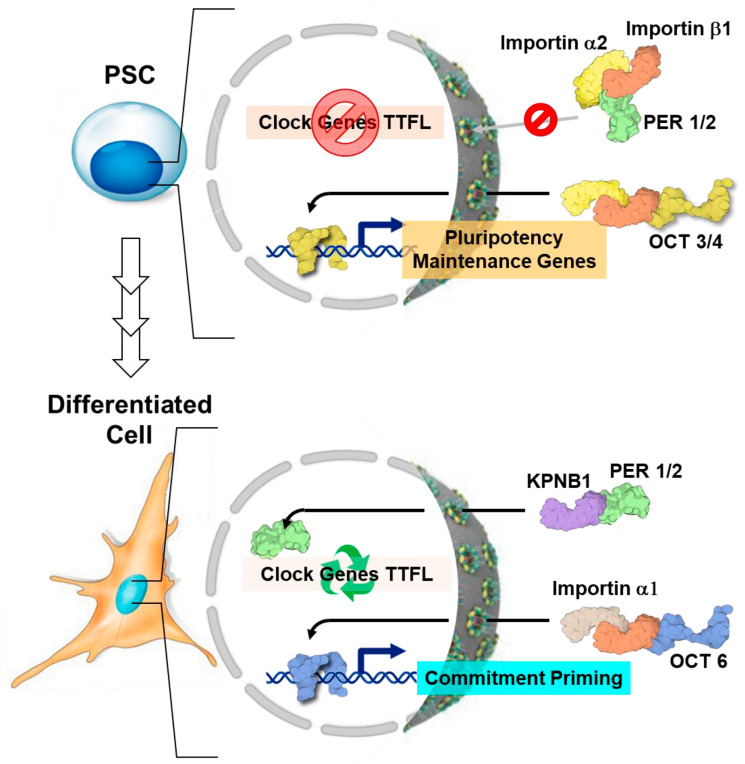
Differential cytosol–nucleus trafficking of circadian factors. Cytoplasmic retention of PER proteins and differentiation factors due to up-expression of importin α2 in PSCs prevents the proper nuclear function of the negative feedback loop required for cyclic circadian regulation and commitment induction (**upper panel**). Alternative expression of KPNB1 and importin α1 induces nuclear localization of core clock proteins along with commitment factors, resulting in establishment of circadian oscillations (**lower panel**). The look of the proteins shown is from the respective PDB depository code and created with BioRender.com. TTFL, transcriptional/translational feedback loops; PSC, pluripotent stem cell. See text for further details.

**Figure 4 ijms-25-02063-f004:**
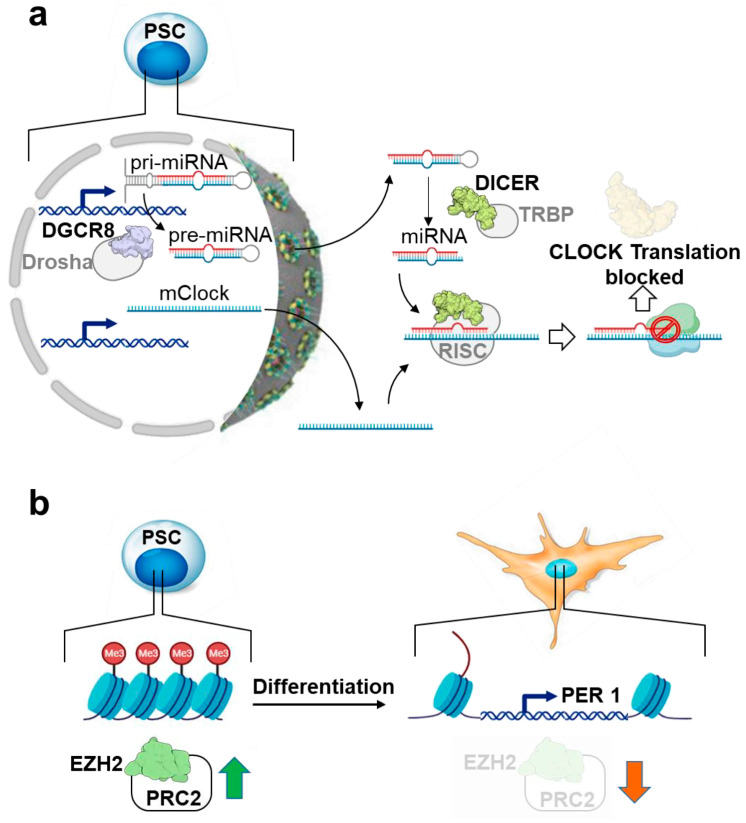
Mechanisms underlying the inhibition of circadian oscillation in pluripotent stem cells. (**a**) Though being expressed at the mRNA level, CLOCK protein translation is almost absent in PSCs because of microRNA (Dicer/DGCR8)-mediated post-transcriptional repression. (**b**) Hypermethylation of histone H3 lysine 27 trimethylation (H3K27me3) in the *Per1* promoter, catalyzed by the histone methyl-transferase EZH2 (enhancer of zest homologue) of polycomb repressive complex 2 (PRC2), is responsible for circadian rhythm repression in PSCs. The image of the proteins shown is from the respective PDB depository code and created with BioRender.com. Green and red arrows stand for up- and down-regulation of EZH2/PRCs, respectively. See text for further details.

**Figure 5 ijms-25-02063-f005:**
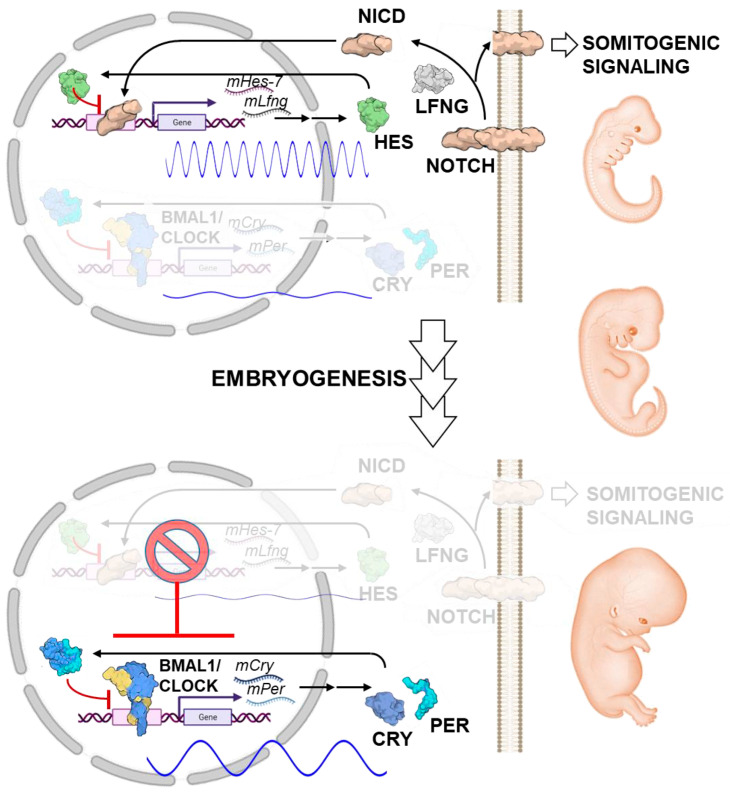
Biological significance of the delayed emergence of circadian clock oscillation in mammalian development. The upper panel shows the early- to middle-developmental stages where the cell-autonomous ultradian rhythm of the somitogenic segmentation clock, driven by a negative feedback loop involving Hes7 oscillation and NOTCH signaling, is essential for a proper developmental process. Expression of the key circadian components CLOCK/BMAL1 (shown in the lower panel) interferes with the Hes7 oscillations-mediated segmentation clock underlying the need of suppressing a functional CLOCK/BMAL1-mediated circadian clockwork for an unharmed process of mammalian embryogenesis to develop. NICD, Notch intracellular domain; LFNG, Beta-1,3-N-acetylglucosaminyltransferase lunatic fringe. The image of the proteins shown is from the respective PDB depository code and created with BioRender.com. See text for further details.

**Figure 6 ijms-25-02063-f006:**
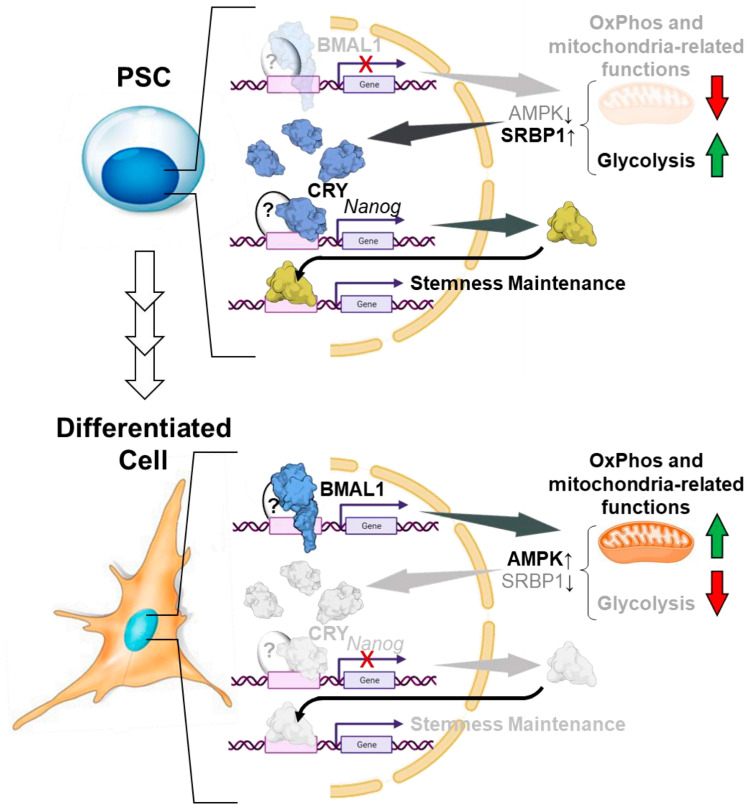
Clock-independent function for circadian clock genes in pluripotent stem cells. The upper panel shows the non-canonical function of CRY1 in controlling the capacity of self-renewal, stemness maintenance, and metabolic programs unique to PSCs. Nuclear accumulation of CRY1 is driven by down-regulation of its AMPK-mediated proteolytic degradation. The low expression of BMAL1, known to positively modulate mitochondrial respiratory functions, contributes to the PSCs’ metabolic signature. The lower panel illustrates the reversal of the above-mentioned PSCs features following induction of their differentiation consequently to down- and up-regulation of CRY1 and BMAL1, respectively. It is highlighted that the transcriptional activity of CRY1 and BMAL1 is exerted on gene responsive elements unrelated to the circadian clockwork. The diagram of the proteins shown is from the respective PDB depository code and created with BioRender.com. Green and red arrows stand for up- and down-regulation of mitochondrial and glycolytic functions, respectively. Question mark indicates unknown co-transcription factor. See text for further details.

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
