# Peer review of "“Time Is out of Joint” in Pluripotent Stem Cells: How and Why"

_ijms, 2024, doi:10.3390/ijms25042063_

Round 1

Reviewer 1 Report

Comments and Suggestions for Authors

This review summarizes how and why circadian oscillations are hampered in Pluripotent Stem cells. In addition, new insights about non-canonical functions of circadian factors in the balance between pluripotency identity and metabolic-driven cell reprogramming are also reviewed. The paper is well organized, and the idea is attractive to a wide audience. Upon a minor correction is resolved, I would like to see its revision published in IJMS.

1.      The English throughout the manuscript needs editing and correction.

2.      The abstract contains information on the manuscript.

3.      The introduction should be improved and needs more literature review. For example Line 36, Page 2, the authors mentioned recent studies “Studies over the last years have revealed the.” but they just referenced 1 paper. I suggest stressing and briefly citing some updated and related papers.

4.      What is the novelty and difference of your review compared to other published manuscripts? The authors of this study need to present more highlights.

5.      Page 2, line 74: The first appearance of CLOCK and BMAL1 in the paper should be after its full name.

6.      Since the authors abbreviated "pluripotent stem cells" as "PSCs", there should not be an excess of "pluripotent stem cells" throughout the manuscript.

7.      Some titles are too long and should be shortened. The obvious examples in this note are Sections 6 and 7.

8.      Page 6, line 239: How did the circadian clock genes play a crucial role in governing the cell cycle of mouse ESCs?? Please explain more.

9.      There is too much background introduction in the Sections 2 and 3.

10.  It should be pointed out that the evidence provided is not enough to support the conclusion in this manuscript. To make the conclusion section more informative, it must be modified to consider future perspectives, discuss the evidence presented in the manuscript, and explain more about the effect of reprogramming on regenerative capacity and the aging process.

11. Figure 1 should be located after the paragraph that mentioned in the manuscript and please remove the background.

12.  Please cite and provide more up-to-date studies.

Comments on the Quality of English Language

The English throughout the manuscript needs editing and correction.

Author Response

We thank the reviewer for the pertinent observations/suggestions. We have carefully evaluated the comments and the manuscript has been revised accordingly and ready for resubmission. Please find attached our point-by-point replies to the concerns raised and the corresponding revisions/corrections in track changes in the re-submitted files

Reviewer 2 Report

Comments and Suggestions for Authors

This review addresses an interesting aspect of stem cell biology. It summarizes and discusses the amazing finding that circadian gene expression in pluripotent stem cells does not show the typical escalating pattern. While the reported findings recurrent and complete, the text lacks focus and clarity.  The introduction to iPSCs does not fit to the highly specialized main aspect. I suggest to shorten the review and to reduce the number and improve some detail of the figures (for example Fig.2). Furthermore, the figures contain elements that clash in style. For example, the grey background obscures the message of Fig. 1. The letters inside the the DNA is obscured by the symbols. The fonts sizes of the smaller writing are illegible. The embryos of Fig. 5 should be improved.

Comments on the Quality of English Language

The text and especially the figures contain grammar and spelling mistakes (manteinance). Colloquialisms should be avoided (isn't).

Author Response

We thank the reviewer for the pertinent observations/suggestions. We have carefully evaluated the comments and the manuscript has been revised accordingly and ready for resubmission. Please find attached our point-by-point replies to the concerns raised and the corresponding revisions/corrections highlighted/in track changes in the re-submitted files

Round 2

Reviewer 2 Report

Comments and Suggestions for Authors

The authors have addressed my concerns appropriately.